# TransForm: Dynamic Format Selection for MCP Tool Outputs in Agentic Data Systems

Shashank Mukkera
Purdue University
USA
smukkera@purdue.edu

Chunwei Liu
Purdue University
USA
chunwei@purdue.edu

## ABSTRACT

Model Context Protocol (MCP) standardizes tool use for AI agents, but its JSON-RPC control plane is a poor fit for large tabular results: repeated keys inflate bytes, stress context budgets, and block incremental consumption. We present TransForm, a control/data-plane split in which MCP returns a small envelope plus a machine-readable descriptor, and large payloads are delivered over streamable HTTP as JSON, Parquet, or Arrow IPC (blob or chunk stream). A target-driven selector chooses the representation per query—minimizing bytes, end-to-end latency, or time-to-first-rows—with fused server-side selection that avoids an extra control-plane exchange. On BIRD dev (1534 gold-SQL queries), TransForm reduces aggregate wire volume $\approx 6\times$ versus JSON baseline (43.4 MB $\rightarrow$ 7.4 MB). On a Java MCP server, TransForm lowers aggregate end-to-end latency by 23% (1.3×) over the full BIRD dev workload while retaining byte savings. Under emulated WAN/Cellular/BadWifi links, TransForm improves p95 and aggregate transport latency on large-result queries by up to 2.0×, with gains growing as bandwidth shrinks.

**VLDB Workshop Reference Format:**

Shashank Mukkera and Chunwei Liu. TransForm: Dynamic Format Selection for MCP Tool Outputs in Agentic Data Systems. VLDB 2026 Workshop: NOVAS.

**VLDB Workshop Artifact Availability:**

The source code, data, and/or other artifacts have been made available at https://github.com/shashank524/TransForm.

## 1 INTRODUCTION

AI agents are increasingly deployed as tool-using systems: they issue queries, call APIs, manipulate files, and iterate over intermediate results in plan–act–reflect loops. In **hybrid relational–AI** workloads—especially text-to-SQL and analytics tools [6, 10]—the dominant cost often shifts from query execution to *moving and representing* tabular tool outputs across the agent loop [2, 9]. Model Context Protocol (MCP) is emerging as an interoperability layer for this ecosystem, standardizing a JSON-RPC control plane with primitives such as tools, resources, prompts, and notifications [3]. MCP resembles a "REST API for AI," but it is fundamentally message-oriented: tool inputs and outputs become part of the agent's working memory and often flow through model context.

**Motivation: large structured tool outputs.** A common pattern is text-to-SQL tools that return relational query results. In today's MCP deployments, these results are frequently encoded directly as JSON objects (e.g., lists of records) and returned through the MCP control plane. JSON is convenient and widely supported, but it is structurally inefficient for tables: it repeats column names, inflates bytes, and forces clients (and models) to receive the entire payload before acting. In agentic workloads, agents often do not need the full table to proceed, and they benefit from time-to-first-rows (TTFR) and the ability to terminate early when confidence is sufficient.

**MCP layering and deployment realities.** MCP consists of (i) a *data layer* that defines JSON-RPC lifecycle and core primitives, and (ii) a *transport layer* that defines how messages flow (e.g., local STDIO vs. remote streamable HTTP). Local MCP servers that use STDIO typically serve a single client process, while remote MCP servers over streamable HTTP are designed to serve many clients and must account for authorization, payload limits, and fairness. These realities motivate a separation between the control plane (small JSON-RPC messages) and a data plane optimized for large artifacts—analogous to how database systems separate query plans from storage/scan formats.

**Our approach.** We present **TransForm**, a dynamic format-selection framework for MCP tool outputs that separates *control* from *data*. Tools return a small MCP envelope with (a) concise natural language and (b) structured descriptors describing how to fetch and decode large results. Large structured outputs are delivered on a streamable HTTP data plane using Parquet as a binary columnar representation, either as a whole blob or as a length-prefixed chunk stream. This design supports incremental client processing, bounded memory, and early termination; it also accommodates Arrow IPC as a decode-speed reference.

**Contributions.**

- **Control/data-plane split for MCP results.** A descriptor-based interface that keeps MCP JSON-RPC messages small while delivering large tabular payloads via streamable HTTP.
- **Parquet blob and chunk streaming.** Whole-blob delivery and incremental, length-prefixed chunk streaming to enable low TTFR and early termination.
- **Dynamic format selection with fused server paths.** Target-driven selection using size hints and optional network+decode proxies, supporting client-side selection, one-shot server-side selection, and SQL-fused inline delivery.
- **Evaluation on controlled and real workloads.** TPC-DS slices characterize size scaling and decode proxies; BIRD dev gold SQL measures payload and latency under localhost and emulated networks, demonstrating $\approx 6\times$ aggregate wire

reduction and Pareto improvements when fused selection is used.

## 2 RELATED WORK

*MCP ecosystem.* MCP standardizes how AI clients call tools and consume results through a JSON-RPC data layer and multiple transports. A recent measurement study highlights that MCP is deployed at scale but remains heterogeneous, with under-adoption of scalable transports in clients [3]. Practitioner guides similarly advocate reference-based delivery for large multimodal payloads (store externally, return a URI or descriptor in the MCP message) rather than embedding bytes in JSON [1]. TransFormcomplements MCP's control plane by introducing a tabular data plane while remaining backward-compatible with JSON-only clients.

*Agent-first data systems.* Recent work argues that LLM agents will become dominant "users" of data systems, exhibiting high throughput, redundancy, and a need for rapid iteration ("agentic speculation") [9]. Our work is orthogonal to query execution optimizations inside the database: we focus on representation and delivery of tool results across the control/data boundary.

*Data formats and adaptive selection.* Columnar formats such as Parquet and Arrow provide strong compression and fast interchange [8]. Byte-level analyses show that text-based encodings can reduce redundancy for tabular structures compared to generic JSON [11]. Data-driven column encoding selection has been studied for analytical storage [5], and adaptive, input-dependent selection among compression methods has been studied for edge devices [7]; TransFormapplies this lens to tool-result delivery, selecting among JSON, Parquet, and streaming formats based on explicit optimization targets and measurable hints. Table-reasoning agents that issue many exploratory queries amplify the importance of efficient tool-result delivery [4, 6].

## 3 MOTIVATION

Three observations motivate adaptive transport rather than JSON-only delivery:

- **Agentic workloads are latency-sensitive.** Result delivery is a first-class bottleneck in plan–act–reflect loops [9].
- **MCP is real but uneven for large structured outputs.** Ecosystem clients and servers vary in how they handle large tool payloads [3].
- **JSON does not scale for tables.** Columnar encodings trade encode cost for wire efficiency and streaming [8, 11].

Figure 1 shows encode+decode latency versus payload size on TPC-DS-derived tables. JSON has the lowest cost at very small outputs (< 3–40 KB JSON-equivalent), while Parquet and Arrow IPC win as size grows. This crossover motivates a selector that falls back to inline JSON for tiny results—the common case on BIRD dev—and activates columnar transport only when transfer savings dominate encode overhead.

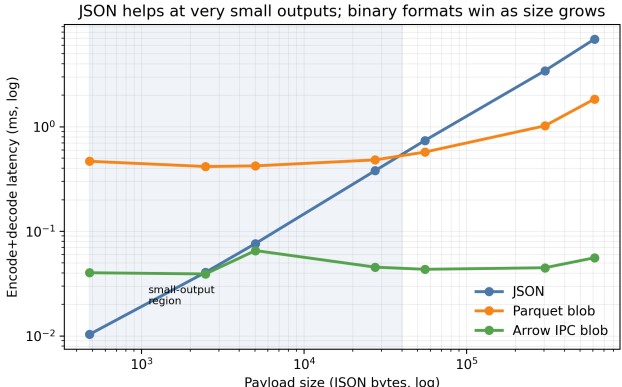

**Figure 1: Encode+decode latency vs. payload size on TPC-DS-derived tables. Each point is one (*rows*, *cols*) slice; JSON is cheapest below ≈3–40 KB JSON-equivalent, while Parquet and Arrow IPC win as tables grow.**

## 4 TRANSFORM DESIGN

### 4.1 Control plane vs. data plane

TransFormaddresses the mismatch between MCP's control plane and large structured payloads by splitting responsibilities:

- **Control plane (MCP JSON-RPC).** Tool calls return a small envelope containing a short natural-language summary and a machine-readable *descriptor* that tells the client how to retrieve and decode the full payload.
- **Data plane (HTTP).** Large payload bytes are delivered over HTTP endpoints as either a whole blob or a length-prefixed stream, keeping MCP messages small and enabling incremental consumption.

### 4.2 Descriptors and transport modes

For tabular payloads, each descriptor specifies the chosen format, transport mode (inline JSON, whole-blob HTTP fetch, or length-prefixed streaming), and encoding (JSON records, Parquet, or Arrow IPC). Unstructured payloads (text, raw, or compressed blobs) follow the same pattern but are out of scope for our evaluation.

### 4.3 Tabular format arms

TransFormsupports multiple arms for large tabular results:

- **JSON records (baseline).** Inline in the MCP response when the table is small; compatibility fallback for legacy clients.
- **Parquet blob.** Whole-file HTTP fetch when minimizing wire bytes matters most.
- **Parquet stream.** Length-prefixed HTTP chunks when time-to-first-rows or early termination matters.
- **Arrow IPC (optional).** Fast in-memory interchange; included as a decode-speed reference, not the BIRD default.

## 4.4 Format selection and placement

Where format selection runs matters as much as *which* format is chosen. Both workflows share the same format arms and optimization targets (wire bytes, end-to-end latency, or time-to-first-rows); they differ in how many control-plane exchanges sit on the critical path.

*Client-driven selection.* After registering a materialized result, the MCP client issues a *second* control-plane call to obtain size hints—approximate row/column counts and byte estimates per encoding. The client runs a deterministic selector locally, then invokes the fetch path for the chosen representation (inline JSON or an HTTP descriptor). This design keeps legacy JSON-only servers usable and lets agent-specific policies live in the client, but every extra MCP request–response pair adds fixed latency: connection setup, JSON-RPC framing, server dispatch, and waiting for the reply. On localhost this overhead is modest ($\approx$1.3 ms for the hinting step in our Java server), yet it is paid on *every* query; over WAN or cellular links it compounds with network round-trip time and can erase the benefit of smaller payloads (Figure 2).

*TransForm (fused server-side).* TransForm performs hinting, selection, and delivery inside one MCP tool call. The server returns either inline JSON (when the table is small) or a compact descriptor (format, transport mode, data-plane URL) when columnar delivery wins. The client may still fetch bytes over HTTP, but it never needs a separate control-plane round trip for hints or format choice. SQL-fused inline goes further by combining query execution with selection in a single call when the server executes the SQL directly.

For latency-oriented targets, selection can incorporate a simple cost model: estimated transfer time under assumed bandwidth plus a decode proxy (nanoseconds per byte, calibrated on TPC-DS slices) [7].

## 4.5 Codec and encoding within Parquet

Within the Parquet arm, compression (Snappy, Zstd, Gzip) and column encodings (default vs. data-driven, inspired by CodecDB [5]) are configurable. Rules include dictionary encoding for low cardinality, delta packing for sorted integers, and plain encoding as fallback. All BIRD transport experiments fix Snappy with default column encodings; codec and encoding strategy are ablated offline on TPC-DS slices (Section 5).

## 4.6 Implementation and caching

We implement TransForm in Python and Java (Spring Boot) with matching MCP tool surfaces. The implementation is organized around the two selection placements in Figure 2: a *split* client-driven path (register, then separate hint and fetch exchanges) and a *fused* server-side path that completes hinting, format choice, and delivery in a single MCP response.

*Round trips and shared delivery.* The baseline and client-driven arms both register a materialized result first, then issue at least one additional MCP call for transport. Client-driven selection adds a second post-registration round trip devoted entirely to size hints before the client issues a fetch for the chosen encoding. TransForm collapses the hint and fetch stages into one fused MCP exchange:

the server reads cached hints, runs the same deterministic selector used on the client, and returns either inline JSON or a descriptor pointing at the data plane. A follow-up HTTP transfer happens only when the selected format is not inline (Figure 2b). Both Python and Java servers route fused delivery through a single shared code path so hint computation is not duplicated across tool entry points.

*Registration cache.* At registration or inline materialization, the server materializes the query result once and pre-computes size hints plus encoded payloads in a bounded in-memory cache (LRU eviction). Cached entries hold tabular views, JSON record lists, and pre-encoded Parquet/IPC bytes where applicable. This cache is what makes fused selection practical: the server can answer a one-shot MCP call without re-reading source files or re-encoding columnar bytes on every hint or fetch. Client-driven hinting benefits from the same cache—hints are cheap to serve—but the client still pays the extra synchronous round trip to request them.

*Lazy vs. eager encoding.* Columnar bytes may be produced eagerly at registration (warming the cache for repeat access) or lazily on the first HTTP blob request. Fused paths prefer eager hints and, when possible, cached encodings so the MCP response can return immediately; lazy encoding defers work until the client actually pulls from the data plane. For small results that stay inline as JSON, neither a second MCP call nor an HTTP fetch is required.

## 5 EVALUATION

### 5.1 Experimental setup

**Servers and protocols.** We run local MCP servers (Python and Java) exposing a JSON-RPC control plane and an HTTP data plane. On BIRD dev, we hold the executed query result fixed and compare three transport arms that differ only in how format is chosen and delivered:

- **Baseline (JSON).** Register a materialized result, then fetch the full table as JSON records—the default MCP compatibility path.
- **Client-driven selection.** Register, then request format hints in a separate MCP call, select locally, and fetch the chosen representation—two post-registration MCP round trips.
- **TransForm.** Register, then issue one fused MCP call in which the server hints, selects, and returns either inline JSON or a data-plane descriptor—one post-registration MCP round trip.

**Datasets.** We use (i) synthetic grids with controlled ($n\_rows$, $n\_cols$), (ii) TPC-DS `catalog_sales` slices [12] to calibrate decode proxies and offline Parquet codec ablations, and (iii) BIRD dev query results (SQLite port) [6] for end-to-end transport evaluation with gold SQL (isolating delivery from NL2SQL quality).

**Network profiles.** For bandwidth-limited experiments we replay the full BIRD workload under four emulated profiles via `tc/netem`: LAN (1 ms, 1 Gbit), WAN (40 ms, 50 Mbit), Cellular (80 ms, 10 Mbit, 0.1% loss), BadWifi (30 ms, 5 Mbit, 1% loss).

**Metrics.** We report wire bytes, per-stage latencies, fair end-to-end totals (register + transport), and tail statistics (p95/p99). On BIRD, medians are often tiny ($\approx$57 bytes); we emphasize aggregate sums and tail metrics where savings concentrate.

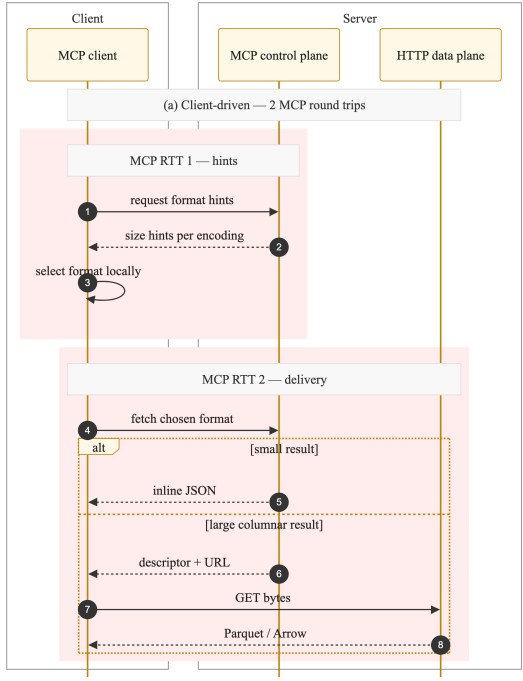
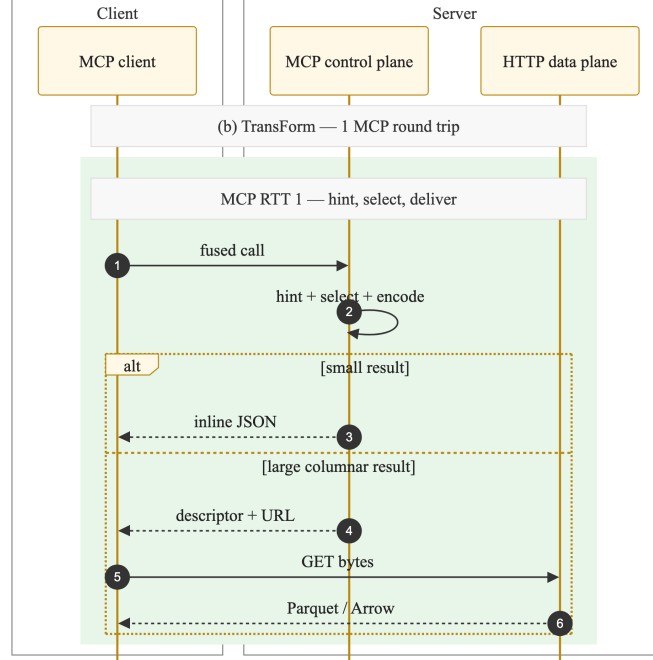

**(a) Client-driven: two MCP round trips.**  **(b) TransForm: one fused MCP round trip.**

**Figure 2: Control-plane exchanges after materialized-result registration (registration omitted). Dashed arrows are MCP replies; large columnar results use the HTTP data plane.**

## 5.2 Streaming, decode proxies, and codecs

Chunk streaming improves responsiveness when the client can begin processing before receiving the full result. On TPC-DS streaming experiments, first-chunk byte volume depends primarily on column count rather than total table size: for a fixed schema, the initial chunk stays constant as rows grow, so TTFR can improve even when the full payload is large. Parquet streams emit smaller first chunks than Arrow IPC at the same column width (roughly 5–8× less at 34 columns in our slices), favoring Parquet when time-to-first-rows matters.

Within Parquet, codec choice trades wire volume against encode+decode CPU. Figure 3 reports bytes at 500K rows and 34 columns across codec/strategy pairs. Stronger compression and data-driven column encodings reduce size—Zstd with data-driven encoding is smallest (≈27 MB) versus ≈36 MB for Snappy default—but Gzip pays a steep CPU penalty: combined encode and decode rises to ≈4 s, roughly an order of magnitude above Snappy (≈0.5 s) at the same slice. Zstd narrows the byte gap without that penalty, yet Snappy remains within a few percent of Zstd on decode time while keeping implementation simple. We therefore fix Snappy for BIRD, where per-query encode cost is on the critical path and most results are small enough that codec differences matter only in the tail.

## 6 BIRD TRANSPORT RESULTS

We evaluate TransFormon BIRD dev with **gold SQL** to isolate transport effects from NL2SQL quality. For register-based arms, the client

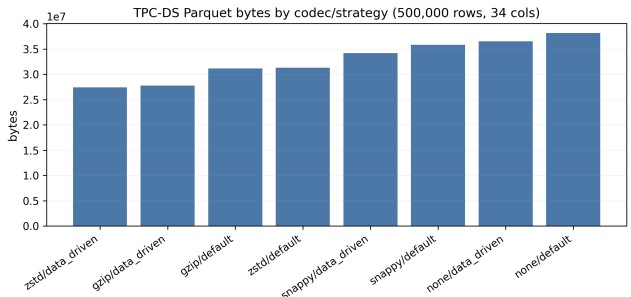

**Figure 3: TPC-DS Parquet wire bytes by compression codec and column-encoding strategy (500K rows, 34 cols). Bars are ordered by increasing size; data-driven encodings (CodecDB-inspired) shave up to ≈20% versus default encodings at the same codec.**

executes reference SQL, materializes once, and compares transport on the same materialized result.

*Payload (wire bytes).* On BIRD full dev (1534 queries), aggregate wire volume decreases from 43.4 MB (baseline JSON) to 7.4 MB under TransForm (≈6×). Medians remain small (45–57 bytes) because most queries return tiny results; savings concentrate on the tail (≈8% of queries choose Parquet/Arrow). Figure 4 compares median wire bytes for baseline JSON versus TransForm across LAN, WAN, Cellular, and BadWifi emulations; gaps widen as link capacity

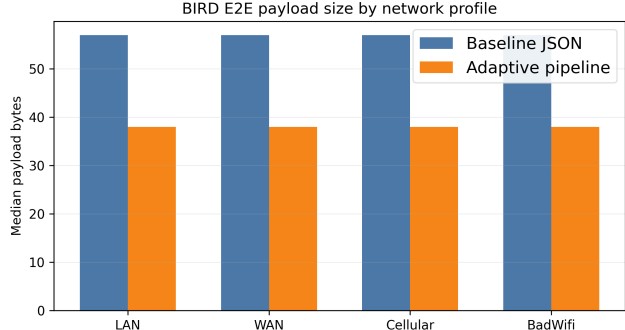

Figure 4: Median wire bytes per query by network profile (baseline JSON vs. TransForm). Savings appear once payloads exceed the JSON crossover in Figure 1.

Table 1: BIRD dev gold (1534 queries): fair end-to-end aggregate latency and aggregate bytes.

| Arm | Aggregate (s) | Sum bytes |
|---|---|---|
| Baseline (JSON) | 8.7 | 43.4 MB |
| Client-driven | 8.8 | 7.4 MB |
| **TransForm** | **6.7** | **7.4 MB** |

shrinks because smaller columnar payloads spend less time on the wire.

*Localhost latency and selection placement.* Client-driven selection achieves the same byte savings as TransForm but pays an additional control-plane round trip for format hints. Summed over all 1534 queries on our Java server, client-driven aggregate latency matches baseline (≈8.7 s) despite smaller per-query payloads—the extra hinting exchange (≈1.3 ms median per query) erases localhost gains from byte reduction. TransForm lowers aggregate latency by 23% versus baseline (1.3×) by fusing hinting, selection, and delivery into one post-registration MCP call instead of two (Table 1).

Per-stage medians confirm where the penalty sits: baseline transport totals ≈60 ms, client-driven totals ≈166 ms (materialize ≈58 ms + describe ≈57 ms + fetch ≈51 ms), and TransForm totals ≈71 ms in a single fused call—near baseline despite smaller payloads. On a per-query scatter over the workload, most BIRD queries cluster at small payloads (<10 KB) and sub-10 ms latency; savings appear in the tail, where JSON payloads exceed ~100 KB and latency rises sharply while TransForm stays on the smaller columnar representation.

*Bandwidth-limited latency.* On localhost, per-query latency is dominated by fixed request–response overhead (JSON-RPC parsing, server scheduling, and synchronous round-trip delay) rather than bytes on the wire, because loopback bandwidth is effectively unlimited. Smaller payloads therefore do not always shorten median latency when results are tiny. Under emulated WAN, cellular, and bad-WiFi profiles, transfer time grows with payload size and shrinks with link capacity; columnar encodings then translate directly into tail-latency wins. On WAN, TransForm reduces p99 latency from 1.63 s to 1.24 s (−24%) and total workload time from 356 s to 318 s

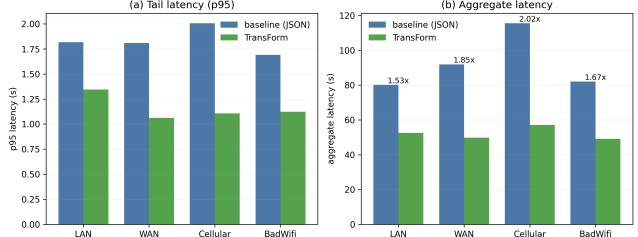

Figure 5: Columnar-selected BIRD queries ($n$=166): baseline JSON vs. TransForm. Left: p95 latency by network profile. Right: aggregate transport time; labels show baseline/TransForm speedup.

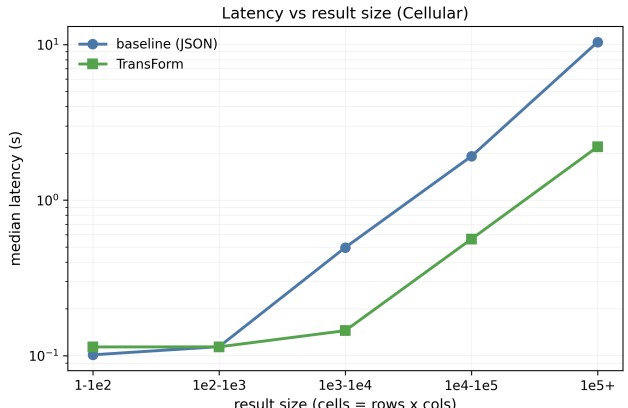

Figure 6: Median transport latency vs. result-size bucket on Cellular. Both arms overlap for small outputs; TransForm pulls ahead as cell count grows.

(1.12×); on cellular, p99 falls from 1.91 s to 1.24 s (1.13× total). Client-driven selection does not share this benefit on small-result queries: it still pays the extra hinting exchange on every query regardless of whether columnar transport is chosen.

Figure 5 focuses on the 166 queries where TransForm chose a columnar format (10.8% of the workload): left, p95 latency by profile; right, aggregate transport time with speedup annotations. Figure 6 plots median latency versus result-size bucket on Cellular—baseline and TransForm coincide for tiny outputs, then diverge past ≈ $10^3$–$10^4$ cells. Figure 7 isolates the selection-placement penalty on WAN: client-driven latency (two MCP round trips) versus TransForm (one round trip) for $n$=1364 paired queries.

Table 2 summarizes the columnar-selected subset across profiles. Aggregate client-driven workload on WAN is +73% over baseline (353 s vs. 204 s) while TransForm stays near baseline (208 s).

## 7 CONCLUSION

TransForm shows that MCP tool outputs benefit from treating format and transport as a first-class, per-query decision. JSON remains the right choice for tiny results; fused server-side selection scales better when transfer cost matters, delivering both byte and latency

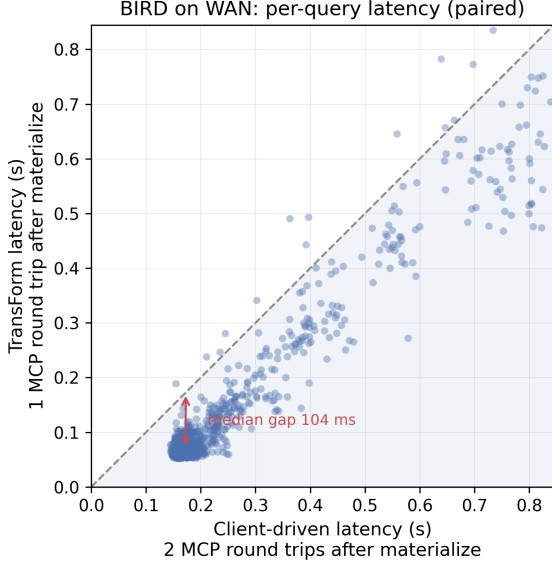

**Figure 7: Paired per-query latency on WAN: client-driven (2 MCP round trips) vs. TransForm (1 round trip). Points below the dashed diagonal are TransForm wins (99%; median speedup 2.4×). The shaded region and red bracket mark the typical gap from the extra hinting round trip (≈104 ms median).**

**Table 2: Bandwidth-limited BIRD: baseline / TransForm on columnar-selected queries (_n_=166).**

| Profile | Median (ms) | p95 (s) | Agg. speedup | Win rate |
|---------|------------|---------|--------------|----------|
| LAN | 79 / 53 | 1.82 / 1.34 | 1.53× | 48% |
| WAN | 131 / 89 | 1.81 / 1.06 | 1.85× | 62% |
| Cellular | 198 / 128 | 2.01 / 1.11 | 2.02× | 65% |
| BadWifi | 131 / 78 | 1.69 / 1.12 | 1.67× | 60% |

wins. **Selection placement** is as important as format choice: pushing hints and selection to the client saves server complexity but adds a control-plane round trip that hurts latency on every query; TransForm retains byte savings without that penalty (Figure 2).

**Scope and limitations.** This work optimizes **tabular** relational tool outputs in hybrid SQL+LLM agent loops; semantic operators, vector RAG, and general multimodal QA are out of scope. Network results use `tc/netem` emulation; live WAN evaluation with competing flows remains future work. Gold SQL isolates transport from NL2SQL quality. Dynamic per-query codec selection, richer multimodal payload conventions, and multi-tenant authorization/auditing on the data plane are open directions. Integrating privacy-preserving transformations and workload-aware adaptive selection policies under agent context budgets is a natural next step.

## ACKNOWLEDGMENTS

This work was supported by the Department of Computer Science at Purdue University.

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
