# OpenReview forum: "TransForm: Dynamic Format Selection for MCP Tool Outputs in Agentic Data Systems"
_VLDB.org/2026/Workshop/NOVAS — NOVAS 2026_

### Official Review · Reviewer_8HsU · 2026-07-02

**Confidence:** 4

**Improvement Opportunities:**

O1: More baseline comparisons [1) compressed JSON and 2) Arrow Flight].
 (a) The 6× byte win is measured only against raw, uncompressed JSON. But gzip over the JSON records: a one-line HTTP content-encoding change, attacking exactly the redundancy you blame and would likely capture much of that 6× without any Parquet or data-plane split. You already use gzip inside the Parquet arm, so adding a compressed JSON column to Table 1 is cheap and would isolate what columnar/Parquet specifically buys
(b) Arrow Flight (gRPC + Arrow IPC record-batch streams over HTTP/2) is the canonical, production-grade, efficient tabular data plane over HTTP and is essentially a more mature instantiation of your control/data split; yet it is never named or compared. You do not need to reimplement it, but you could discuss it and state your differentiator (MCP-native envelope + JSON fallback + fused per-query selection)

O2: The agentic claim could be softened since there is no evaluation.
The paper is framed around agent loops and context budgets, but the evaluation never measures either. # of tokens in the agentic world is never evaluated as the metric; every metric is wire bytes or transport latency. And note that Parquet/Arrow bytes never enter the model context at all --- they are fetched over HTTP --- so the 6× wire reduction is largely orthogonal to the context-budget claim. The real context saving is the small descriptor vs a big inline JSON, which you could measure in tokens (descriptor vs raw-JSON) cheaply. Similarly, TTFR and early termination are headline features, but the only evidence is that first-chunk bytes stay constant as rows grow; no experiment shows an agent actually terminating early or completing a task faster. Either add a token measurement and a small early-termination demonstration, or explicitly reframe as a transport-efficiency paper and move the agentic claims to motivation-only.

O3: Evaluate the selector.
The thesis is a target-driven selector, but its decision quality is never measured. There is no oracle / best-possible-format comparison and no "how often did it pick the optimal representation," so a reader can not tell whether the wins come from the selector choosing well or simply from the fact that any columnar format beats raw JSON at large sizes. This matters because Table 2's LAN row shows a  48% win rate on the low-latency link

**Minor Comments:**

- hardware spec could be stated somewhere

- "TransForm" is glued to the next word throughout ("TransFormcomplements", "TransFormapplies", "We implement TransFormin", "We evaluate TransFormon")... Looks like a LaTeX macro missing its trailing space

**Short Summary:**

The paper tackles a real and timely pain point: the Model Context Protocol (MCP) is becoming the standard way AI agents call tools, but its JSON-RPC control plane is a bad way to move large tabular results. JSON repeats column names, inflates bytes, and forces the client (and the model) to take the whole payload at once. TransForm's answer is a control/data-plane split borrowed from database systems: the MCP call returns a small envelope plus a machine-readable descriptor (what format, how to fetch, how to decode), and the actual bytes go over a streamable HTTP data plane as JSON, Parquet, or Arrow IPC; either a whole blob or a length-prefixed chunk stream. A target-driven selector picks the representation per query to minimize wire bytes, end-to-end latency, or time-to-first-rows.

The part I find good is the placement insight: it is not just which format you pick, but where you pick it. A client-driven design needs an extra control-plane round trip to ask the server for size hints before choosing, and that round trip is paid on every query --- which wipes out the byte savings, especially over WAN. TransForm instead fuses hinting + selection + delivery into one server-side call. A registration cache pre-computes hints and encoded payloads so this one-shot call is cheap.

The evaluation uses TPC-DS slices (to characterize the JSON -> columnar crossover and to ablate codecs offline) and BIRD dev (1534 gold-SQL queries) for transport, under four tc/netem network profiles (LAN/WAN/Cellular/BadWifi). Aggregate wire volume 43.4 MB -> 7.4 MB (~6×); aggregate end-to-end latency −23% on the Java server; and up to ~2× tail-latency improvement on large-result queries under constrained links, with gains growing as bandwidth shrinks.

And artifacts are available.

**Strong Points:**

S1: Solve a real-world problem with clean motivations. MCP-over-JSON for large tabular tool outputs is a genuine and current bottleneck, and the control/data-plane framing is the right lens. The paper is refreshingly candid that most BIRD queries are tiny (median 45–57 bytes), that only ~11% of queries ever go columnar, and that JSON stays the right choice for small results, so it explicitly steers you to aggregate and tail metrics instead of pretending every query wins. The Scope & Limitations section is unusually upfront (emulation vs live WAN, gold-SQL isolation, static codec, multi-tenant auth as future work)

S2: Real contribution/novel idea for the section placement. The per-stage breakdown (client-driven ~166 ms = materialize 58 + describe 57 + fetch 51, vs TransForm ~71 ms in one fused call) and the paired WAN scatter (Fig 7) directly support it. I would point out this to the headline contribution and demote the (borrowed) control/data split

S3: Open-source with strong artifacts. A working Python and Java implementation with an open-source artifact is real and helps reproducibility and further development

---

### Official Review · Reviewer_KGGH · 2026-07-09

**Confidence:** 3

**Improvement Opportunities:**

1. Execution times are reported only post-registration, so the pre-registration cost of materializing and eagerly encoding the columnar payload is never shown. Therefore, it is not evident whether this overhead is justified by later gains.
2. TransForm's total per-query latency slightly exceeds baseline JSON on small results (as discussed in Section 6,  "Per-stage medians confirm where the penalty sits: baseline transport totals ≈60 ms .... and TransForm totals ≈71 ms"). What is the source of this gap?
3. Evaluation might benefit from repeated runs to report variance among different runs.

**Minor Comments:**

1. 'TransForm' is never followed by a whitespace in the text.

**Short Summary:**

The paper introduces TransForm, a method that splits the control and data planes for MCP tool outputs, adjusting format delivery of tabular data results. Besides selecting which representation is most efficient, the other contribution of the paper is showing that where this selection happens matters as much as the choice itself. Results show that picking the right representation contributes to significant byte reduction, yet latency gains are evident only when the selection is fused into a single server-side call.

**Strong Points:**

1. Splitting control/data planes seems a reasonable choice that can potentially lead to considerable byte savings.
2. Pointing out where the format is decided is as important as which format is picked shows that the authors have a good understanding of agent system latency.
3. The future directions that the authors have laid out are sensible and actionable, and they follow naturally from the current prototype.

---

### Official Review · Reviewer_SPHr · 2026-07-10

**Confidence:** 3

**Improvement Opportunities:**

1.The paper could better clarify the generality of the design beyond SQL-style tabular outputs. The motivation is framed around MCP tool outputs broadly, but most of the concrete design and evaluation target relational result tables.
2.The selection policy could be specified more rigorously. The paper mentions goals such as minimizing bytes, end-to-end latency, and time-to-first-rows, but the exact decision logic, thresholds, and cost model are not always fully transparent.
3.The workload distribution is highly skewed toward small results, and only a minority of BIRD queries benefit from columnar formats. The paper already notes this, but the implications could be discussed more directly: TransForm may be most useful for analytic or exploration-heavy workloads, while its benefit for typical short-result tool calls may be limited.

**Minor Comments:**

The evaluation would be stronger with real distributed deployment results.

**Short Summary:**

The paper proposes TransForm, a dynamic format selection framework for MCP tool outputs in agentic data systems. The central idea is to separate the MCP JSON-RPC control plane from the data plane for large structured outputs, returning compact descriptors through MCP while delivering large tabular payloads through HTTP. The paper argues that JSON is convenient but inefficient for large relational results, and shows that adaptive selection between inline JSON, Parquet blob, and Parquet streaming can reduce wire volume and improve latency, especially under bandwidth-constrained settings.

**Strong Points:**

1.The paper identifies a timely and practically relevant problem in agentic data systems. The paper’s focus on the control/data plane mismatch is well motivated.
2.The proposed design is clean and system-oriented. Separating a small MCP envelope from a streamable data plane is a natural but useful abstraction. The descriptor-based approach also seems compatible with existing MCP-style interactions while allowing more efficient encodings for large tabular outputs.
3.The paper does not merely argue that Parquet is smaller than JSON; it studies where format selection should happen.
4.The reported gains are meaningful in the right regimes: roughly 6× aggregate wire-volume reduction and improved latency under constrained network profiles. The paper also acknowledges that many BIRD queries are small, where JSON remains appropriate, which makes the claims more credible.

---

### Decision · Program_Chairs · 2026-07-16

**Decision:**

Accept

**Comment:**

TransForm addresses an important bottleneck in agentic data systems by separating MCP control messages from large tabular payloads and dynamically selecting an efficient transfer format. Its clean system design, strong implementation, and practical findings on selection placement, wire volume, and latency make it highly relevant to the workshop. We hope the paper stimulates further discussion on efficient data movement in agent-tool interactions.